# Fine-Dust-Induced Skin Inflammation: Low-Molecular-Weight Fucoidan Protects Keratinocytes and Underlying Fibroblasts in an Integrated Culture Model

**DOI:** 10.3390/md21010012

**Published:** 2022-12-23

**Authors:** Kirinde Gedara Isuru Sandanuwan Kirindage, Arachchige Maheshika Kumari Jayasinghe, Namki Cho, Seok Ho Cho, Hee Min Yoo, Ilekuttige Priyan Shanura Fernando, Ginnae Ahn

**Affiliations:** 1Department of Food Technology and Nutrition, Chonnam National University, Yeosu 59626, Republic of Korea; 2College of Pharmacy, Chonnam National University, Gwangju 61186, Republic of Korea; 3Department of Clothing and Textiles, Chonnam National University, Gwangju 61186, Republic of Korea; 4Microbiological Analysis Team, Biometrology Group, Korea Research Institute of Standards and Science, Daejeon 34113, Republic of Korea; 5Department of Agricultural, Food and Nutritional Science, University of Alberta, 4-10 Ag/For Building, Edmonton, AB T6G 2P5, Canada

**Keywords:** fine dust, inflammation, HaCaT–HDF integration, fucoidan, NF-κB/MAPK signaling, extra cellular matrix

## Abstract

Prolonged exposure to fine dust (FD) increases the risk of skin inflammation. Stimulated epidermal cells release growth factors into their extracellular environment, which can induce inflammation in dermal cells. Algae are considered rich sources of bioactive materials. The present study emphasized the effect of low-molecular-weight fucoidan isolated from *Sargassum confusum* (LMF) against FD-induced inflammation in HaCaT keratinocytes and underneath fibroblasts (HDFs) in an integrated culture model. HDFs were treated with media from FD-stimulated HaCaT with LMF treatments (preconditioned media). The results suggested that FD increased the oxidative stress in HaCaT, thereby increasing the sub-G_1_ phase of the cell cycle up to 587%, as revealed via flow cytometric analysis. With preconditioned media, HDFs also displayed oxidative stress; however, the increase in the sub-G_1_ phase was insignificant compared with HaCaT. LMF dose-dependently regulated the NF-κB/MAPK signaling in HaCaT. Furthermore, significant downregulation in NF-κB/MAPK signaling, as well as inflammatory cytokines, tissue inhibitors of metalloproteinases, matrix metalloproteinases, and reduction in relative elastase and collagenase activities related to the extracellular matrix degeneration were observed in HDFs with a preconditioned media treatment. Therefore, we concluded that HDFs were protected from inflammation by preconditioned media. Continued research on tissue culture and in vivo studies may reveal the therapeutic potential of LMF.

## 1. Introduction

As the world moves toward the fourth industrial revolution, environmental pollution, and related health concerns are being drastically increased. Among environmental pollutants, airborne particles/fine dust (FD) have detrimental effects on human health and other organisms due to their complexity, which consists of inorganic and organic matter with a variety of masses, sizes, and chemical compositions [1]. Apart from this, air quality indexes for nitric oxide, sulfur dioxide, and carbon monoxide have been scaled to alert the public about real-time air quality [2]. Recent epidemiological data reveal that most common skin diseases, including acne, hyperpigmentation, atopic dermatitis, and psoriasis, are caused by air pollution [3]. The global awareness of FD is much more intense in the East Asian region, particularly in China, Korea, and Japan, owing to the arid or semiarid highlands of Northern China and Mongolia, which contributes to the generation of natural airborne particles on large scale [4]. As reported, airborne FD has around 10 μm of aerodynamic diameter [5]. Even though previous studies mentioned that the impact of FD on skin health is the most debated matter, research revealed that it could penetrate the respiratory tract [1,6]. Long-term exposure to FD could cause complications, such as allergic reactions and inflammatory responses in macrophage cells. Alongside systemic toxicity, cardiac dysfunctions, aggravated asthma, pregnancy complications, and congenital disabilities were identified as results of air pollution [7,8,9].

The human skin is the largest organ in the body and is important in sensation, insulation, the regulation of water retention/loss and temperature, synthesis of vitamin D, and protection of vitamin B folates [10]. Additionally, skin plays a critical role in immunity, where it directly interfaces with the outer environment and tends to get injured by stimuli [10,11]. The skin consists of two major layers: the epidermis and the dermis. The epidermis contains keratinocytes as predominant cells, melanocytes, Langerhans cells, and corneocytes, and has four sub-layers, namely, the stratum corneum, stratum granulosum, stratum spinosum, and stratum basale. The dermis is a more resilient layer that contains fibroblasts, ground substances, and fibers [12].

Skin inflammation is one of the important protective mechanisms that take place against stimuli. However, in certain conditions, such as uncontrolled oxidative stress in cells, inflammation could become a risk factor for numerous chronic diseases [13]. Early studies found that FD stimulates keratinocytes, producing inflammation and skin barrier dysfunction [14,15,16]. As identified and mentioned, exposure to FD increases intracellular reactive oxygen species (ROS) in keratinocytes. Dysregulated ROS production activates the nuclear factor kappa B (NF-κB) signaling and the mitogen-activated protein kinase (MAPK) signaling pathways toward inflammatory reactions; hence, these pathways produce inflammatory cytokines and chemokines [13]. As a result of the cytokine secretions from inflamed keratinocytes, the underlying intercellular environment, as well as human dermal fibroblasts (HDFs), get affected and thereby experience increased oxidative stress. Because of this, the degradation of connective tissues in the dermal layer takes place following the production of matrix metalloproteinases (MMPs) [13].

Naturally available bioactive compounds that are isolated/extracted from marine algae, such as polysaccharides, phenols, sterols, proteins, and a variety of pigments, are recently acquiring interest as pharmaceutical/cosmeceutical agents in related research fields [2,13,15,16,17,18]. A crude methanol extract of edible brown algae *Sargassum confusum* indicated protective effects on β-amyloid-protein-induced HT-22 mouse neuronal cells [19] and phorbol 12-myristate 13-acetate induced mouse ear edema and erythema [20] in previous studies. Among the wide range of identified bioactive agents, fucoidan, which is a sulfated polysaccharide, was known to be an excellent antioxidant, anti-inflammatory, anti-obesity, anticancer, antitumor, antiviral, and antidiabetic agent in many former studies [17]. Features and biological activities of a particular form of fucoidan notably depend on the source, composition, molecular weight, and purity, as well as its structure [17]. Anti-inflammatory activities of low-molecular-weight fucoidan isolated from brown seaweeds on keratinocytes as well as HDFs were investigated in vitro in previous monoculture studies [14,17,21].

The present study investigated the anti-inflammatory effect of low-molecular-weight fucoidan isolated from *S. confusum* (abbreviated as LMF from here onward) in FD-stimulated HaCaT keratinocytes and followed HDFs induced by the fucoidan-primed culture media from FD-stimulated HaCaT keratinocytes (preconditioned culture media) in an integrated culture model. In the integrated culture model, HaCaT keratinocytes and HDFs were cultured separately. The HaCaT keratinocytes, which were stimulated by using FD followed by LMF treatments, were investigated for inflammatory reactions, and media were collected from each experimental group. HDFs were treated with the media collected from HaCaT keratinocytes and investigated for inflammatory reactions according to the experimental design. The study was designed by hypothesizing that keratinocytes, which are the outermost living cells, are susceptible to FD exposure, they may release signaling molecules into their microenvironment, and LMF doses will attenuate the effect of FD stimulation. Those signaling molecules stimulate the underneath HDFs to inflammatory reactions by activating inflammatory signaling pathways, whereas HaCaT keratinocyte and HDF integration mimic the cutaneous tissues.

## 2. Results

### 2.1. LMF Effectively Increased Cell Viability and Abated Oxidative Stress in the Integrated-Culture Model

Cell viability and intracellular ROS production in this study were investigated using a 3-(4,5-dimethylthiazol-2-yl)-2,5-diphenyltetrazolium bromide (MTT) assay and a 2′,7′-dichlorodihydrofluorescein diacetate (DCF-DA) assay, respectively. The levels of fluorescence emission and absorbances were measured by using a SpectraMax M2 microplate reader (Molecular Devices, Silicon Valley, CA, USA). According to the results obtained from the MTT and DCF-DA assays, FD stimulation significantly decreased the cell viability in HaCaT keratinocytes (Figure 1A) and increased intracellular ROS production (Figure 1B). The LMF treatments (31.3, 62.5, and 125 μg/mL) significantly increased the cell viability while decreasing the intracellular ROS production in a dose-dependent manner. Further analysis in the integrated-culture model was conducted to figure out the effect of the FD stimulation and fucoidan treatment on HaCaT keratinocytes in their microenvironment and underneath HDFs. The cell culture media collected from FD-stimulated HaCaT keratinocytes (stimulated culture media) moderately decreased the cell viability and increased the intracellular ROS production compared with the HDF control. The preconditioned culture media promptly recovered the cell viability (Figure 1D) and abated the oxidative stress in the HDFs with LMF doses, as indicated in Figure 1E. Inhibition of intracellular ROS production with LMF treatment in HaCaT keratinocytes and HDFs was further clarified using DCF-DA flow cytometric analysis conducted by using a CytoFLEX flow cytometer (Beckman Coulter, Brea, CA, USA). As indicated in Figure 1C, the FD stimulation shifted the peak to the right, which was reduced with the LMF doses. HDFs with stimulated culture media shifted the peak to the right while preconditioned culture media recovered the shift with the LMF doses (Figure 1F). Collectively, these results conferred the effect of FD stimulation on intracellular ROS production in HaCaT keratinocytes and the effect of stimulated culture media on ROS production in the underlying HDFs. Furthermore, the protective effect of LMF against FD-stimulated HaCaT keratinocytes and preconditioned media-treated HDFs were confirmed by the abovementioned results.

### 2.2. LMF Suppressed Apoptosis in FD-Stimulated HaCaT Keratinocytes and HDFs in the Integrated-Culture Model

The accumulation and appearance of cell cycle peaks related to the sub-G_1_ phase are considered biomarkers for the presence of DNA damage and apoptosis [22]. As per the results obtained from the flow cytometry with a propidium iodide (PI) stain, a significant increase in apoptotic cells was identified from the analysis of sub-G_1_ cells in FD-stimulated HaCaT keratinocytes, which decreased with LMF treatment in a dose-dependent manner (Figure 2A). FD stimulation increased the sub-G_1_ phase up to 587% compared with the sub-G_1_ cells in the control group. LMF treatments of 31.3, 62.5, and 125 μg/mL doses decreased the sub-G_1_ cells to 467%, 259%, and 140%, respectively. As shown in Figure 2B, the cell culture media collected from the HaCaT keratinocyte control group increased the sub-G_1_ cells to 126%. The treatment of the stimulated culture media with HDFs increased the apoptotic cell population at a significantly minor level compared with the change in the sub-G_1_ phase of stimulated culture media-treated HDFs, which was 197%. Preconditioned media decreased the number of cells of the sub-G_1_ phase in HDFs to 149%, 131%, and 124% with the LMF doses, respectively.

### 2.3. LMF Effectively Downregulated the NF-κB/MAPK Signaling in FD-Stimulated HaCaT Keratinocytes and HDFs in the Integrated-Culture Model

NF-κB and MAPK signaling are critical interconnected pathways that can be identified in both keratinocytes and HDFs that are responsible for inflammatory reactions [13,17]. There were significant increases in the phosphorylation of IκBα and NF-κB p65 and the nuclear translocation of NF-κB p65 was observed using Western blot analysis in FD-stimulated HaCaT keratinocytes (Figure 3A). Similarly, as indicated in Figure 3B, the phosphorylation of p38, ERK, and JNK in MAPK signaling was increased in FD-stimulated HaCaT keratinocytes. The LMF treatment significantly and dose-dependently downregulated the NF-κB and MAPK signaling in the stimulated HaCaT keratinocytes (Figure 3A,B). The phosphorylation of the NF-κB/MAPK signaling pathway proteins in stimulated culture media-treated HDFs was significantly increased. Meanwhile, downregulation in the phosphorylation of the IκBα, NF-κB p65, p38, ERK, and JNK was indicated to correspond to LMF treatments in preconditioned media (Figure 4A,B).

### 2.4. Preconditioned Media Downregulated Inflammatory Mediators in HDFs in the Integrated-Culture Model

The mRNA expression of inflammatory mediators was investigated by using reverse transcription–polymerase chain reaction (RT-PCR) analysis. According to the findings, the mRNA expression of inflammatory mediators in stimulated culture media-treated HDFs was significantly higher than in the control group and followed the HaCaT keratinocytes control media-treated group (group 1). The preconditioned media treatment downregulated the levels of interleukin (IL)-1β, -6, -8, -33, and TNF-α in HDFs corresponding to the concentration of LMF (Figure 5A). Furthermore, ELISA analysis of IL-6 and TNF-α confirmed the regulatory effect of LMF on the production of inflammatory mediators in preconditioned media-treated HDFs (Figure 5B).

### 2.5. Preconditioned Media Downregulated MMPs and TIMPs, and Suppressed Collagenase and Elastase Activity in HDFs in the Integrated-Culture Model

As denoted in Figure 6A, the mRNA expression of MMPs that related to the connective tissue degradations in stimulated culture media-treated HDFs was significantly higher than in the control group. The preconditioned media treatment downregulated the levels of MMP1, MMP2, MMP3, MMP8, MMP9, and MMP13 in HDFs corresponding to the concentration of LMF. TIMPs are critical to enabling MMP activities in fibroblasts, and the equilibrium of TIMPs and MMPs is crucial for the physiological functions of the extracellular matrix (ECM) [6]. As per the analysis, increased levels of the expression in TIMP1 and TIMP2 were dose-dependently downregulated by the LMF concentrations in the preconditioned media (Figure 6A). Collagenase and elastase were drastically related to ECM degradation. The increased levels of relative collagenase and elastase activity percentages were suppressed following the LMF treatments in the preconditioned media.

## 3. Discussion

Marine biomass is one of the many types of microorganisms, plants, and animals that make up the total amount of biomass on Earth. Since many millennia ago, there has existed a long tradition of consuming seafood as a delicacy as part of the human diet. Many ancient cultures throughout the world would have thought that marine functional ingredients may contribute to steadily increased longevity, as well as long-term health impacts [23]. Even so, marine biomass is a relatively underutilized resource that has received a lot of attention recently, in part due to the overexploitation of land and the fact that the oceans make up more than 70% of the planet’s surface [24]. With the rise of the so-called blue revolution, the discovery of bioactive constituents from marine sources has been a key focus in many research groups for a long time [24,25]. Algae have been recognized as significant sources of bioactive natural compounds since advancements were presented in related research studies. In particular, the growing trend toward the use of marine natural materials in cosmetics will give the cosmeceutical sector a significant boost due to the functional properties of their bioactive constituents. Meanwhile, cosmeceuticals that incorporate biocompatible natural components are increasingly popular in modern society [26].

Among a vast array of bioactive materials, fucoidan can be effectively extracted from brown algae, such as *Sargassum* spp., *Fucus* spp., *Laminaria japonica*, and *Ecklonia cava* due to its higher reproductive rates than other species, such as *Apostichopus japonicus*, *Holothuria tubulosa*, and *Stichopus japonicus* [27]. The sulfated fucose backbone is the key structure of the fucoidans while consisting of mono-sugars, such as galactose, mannose, glucose, and arabinose, collectively ranging from 40 kDa to 1600 kDa in molecular weight [28,29]. Even though several factors such as the degree of sulphation, substitution groups and their positions, type of sugar, and glycosidation branching determine the antioxidant capacity, the molecular weight significantly influences the radical scavenging activity and reduces the capacity of fucoidan [30]. Crossing the lipid bilayer and exerting its biological activity is limited to the high-molecular-weight fucoidan. The low-molecular-weight fucoidan and its derivatives show high antioxidant capacity [31]. Although some studies reported that the functional properties of sulfated polysaccharides failed to characterize the structural properties [28], fucoidan refined from marine algae was widely studied in in vitro skin care studies [14,17,32,33]. Specifically, the anti-inflammatory effect of fucoidan on FD-stimulated epidermal cells [14], the mixture of TNF-α and IFN-γ-stimulated epidermal cells [17], and the antioxidative effect of fucoidan on UVB-stimulated epidermal cells [32,33] were reported. The anti-inflammatory mechanism of fucoidan via cyclooxygenase 2 (COX2) inhibition [34], NF-κB signaling inhibition in the Wistar rat models [35], MAPK signaling inhibition in cerebral ischemia-reperfusion injured rat model [36], and regulating chemokines and cytokines secretion [17] were identified in recent studies.

FD has the potential to increase oxidative stress and thereby induce inflammatory reactions in various cells, as well as animals. The FD (NIES CRM No.28) used in this study was obtained from the ventilation systems in urban buildings in Beijing and its composition was reported by Mori et al. (2008) [37]. The reported particle sizes ranged from 2.5 μm to 15 μm, with 2.5 μm being the most common size. Fernando et al. (2018) revealed the elemental composition of CRM No.18 by using detailed spectroscopic analyses. From the results, the authors reported that the particles were composed mainly of Ca, Al, Fe, Mg, and polycyclic aromatic hydrocarbons, such as fuoroanthene, pyrene, benzo[a]anthracene, benzo[b]fuoroanthene, benzo[k]fuoroanthene, benzo[a]pyrene, benzo[ghi]perylene, and indeno [1,2,3-cd]pyrene [38]. CRM No.28 exposure promptly increases the intracellular ROS levels and thereby increases the inflammatory reactions, including the activation of NF-κB/MAPK signaling [2]. In the present study, the anti-inflammatory effect of LMF on FD-stimulated HaCaT keratinocytes and thereby preconditioned media-induced effects on HDFs in an integrated culture model were investigated. The previous study revealed that the molecular weight distribution was around 20 kDa in LMF. The study reported that the sulfate content was 23.62% and the total polysaccharide content was 44.9%. The fucose content, galactose content, and glucose content were 64.64%, 5.39%, and 29.97%, respectively [39]. The study was conducted by hypothesizing that dysregulated inflammatory mediators released by HaCaT keratinocytes to their microenvironment can increase the inflammatory reactions in HDFs. As per the result obtained from the cell viability analysis using an MTT assay, LMF significantly increased the cell viability in FD-stimulated HaCaT keratinocytes. The DCF-DA fluorometry and DCF-DA flow cytometric analysis presented in Figure 1 confirmed the significant intracellular ROS inhibition in FD-stimulated HaCaT keratinocytes. Interestingly, the cell viability of the HDFs group that was treated with media collected from the HaCaT keratinocyte control group was increased by up to 116% compared with the HDF control. Further, HDFs indicated a slight increase in intracellular ROS levels. Preconditioned media significantly decreased the intracellular ROS levels in HDFs in incubation with the stimulated culture media. These results collectively confirmed that the risk of exposure to FD can increase the oxidative stress in keratinocytes, and even if FD is unable to reach the dermal layers, changes in the microenvironment in epidermal cells could increase the oxidative stress in HDFs. Similarly, the ability of LMF treatment on epidermal cells to attenuate oxidative stress in epidermal cells and underneath dermal cells was confirmed. In order to understand how LMF recovers cells from toxicity after FD stimulation, the cell cycle was investigated, where regular exposure to oxidative stress led the cell toward apoptosis [40]. For this investigation, cells were stained with PI followed by flow cytometric analysis to monitor the cell cycle. As indicated in Figure 2A, apoptosis progress was observed with the FD stimulation in HaCaT keratinocytes by the increase in the number of sub-G_1_ phase cells. Changes in the sub-G_1_ phase of stimulated culture media and preconditioned media-treated HDF groups, which are illustrated in Figure 2B, were insignificant compared with the results related to the sub-G_1_ phase of FD-stimulated HaCaT keratinocytes. This finding suggested that, even after the increase in ROS by FD exposure to keratinocytes, it had a low risk of causing apoptosis in the underlying HDFs. According to previous studies, alteration in the morphological changes protect the HDFs after incubation with preconditioned media despite the intracellular ROS generation [6].

Then, further experiments were planned to investigate the anti-inflammatory effect of LMF in FD-stimulated HaCaT keratinocytes and underneath HDFs in an integrated culture model. Even though inflammatory responses initiate the healing process against harmful damage by stimuli, uncontrolled inflammation causes chronic diseases [13]. Some of the renowned receptors that initiate inflammatory responses trigger the activation of key intracellular upstream signaling molecules, such as NF-κB and MAPK, which are responsible for inflammatory reactions in cells [16]. Other than that, increased oxidative stress can activate NF-κB signaling [13]. The initiation of inflammation-mediated pathways in FD-stimulated HaCaT keratinocytes was confirmed by the results obtained from a Western blot analysis of NF-κB and MAPK signaling. Phosphorylation of NF-κB p65 and IκBα, while the nuclear translocation of NF-κB p65 was significantly downregulated by LMF treatment in a dose-dependent manner in the FD-stimulated HaCaT keratinocytes (Figure 3). Parallel to the above, changes in NF-κB and MAPK signaling in preconditioned media-treated HDFs were downregulated with the LMF doses (Figure 4).

The production of pro-inflammatory cytokines and chemokines is a fundamental adaptation of the immune response, yet it plays an important role in modulating the structural integrity of the skin’s extracellular matrix (ECM) by regulating MMPs, collagenase, and elastase transcription in HDFs [13]. In recent years, much attention has been paid to the search for compounds that can inhibit MMP activation as an approach to prevent skin aging because of the importance of MMPs in chronic inflammation, wrinkling, arthritis, osteoporosis, periodontal disease, and tumor invasion characterized by excessive ECM degradation in the extracellular space [6]. The significant increases in mRNA expressions of inflammatory cytokines and chemokines (IL-1β, IL-6, IL-8, IL-33, and TNF-α) were observed in FD-stimulated HaCaT-keratinocyte-media-treated HDFs while a preconditioned media treatment effectively downregulated the expression levels in RT-PCR analysis. As per the evidence of several studies that linked IL-6 and TNF-α in the regulation of TIMP1, MMP1, and MMP9 mRNA expression, the levels of mRNA expressions of MMP1, MMP2, MMP3, MMP8, MMP9, MMP13, TIMP1, and TIMP2 were investigated in HDFs. All tested mRNA expression levels of MMP and TIMP were increased with the stimulated media and downregulated by the preconditioned media, as expected. Moreover, a notable decrease in relative collagenase and elastase activities was observed in the HDFs that were subjected to incubation with preconditioned media. Preconditioned media reduced the ECM in HDFs, and the translational investigation of TIMP and MMP levels and collagenase and elastase activities provided a firm conclusion on the ECM-degrading effects of FD. In a nutshell, all results suggested that exposure to FD increased oxidative stress in epidermal cells, thereby increasing apoptosis. Due to changes in the microenvironment of keratinocytes, HDFs also experience oxidative stress but, interestingly, apoptosis in HDFs was insignificant compared with the keratinocytes. The LMF treatment in keratinocytes protected the cells from inflammation and thereby HDF cells were also protected through changes in their microenvironment.

## 4. Materials and Methods

### 4.1. Materials

HaCaT keratinocytes and human dermal fibroblasts were purchased from the Korean Cell Line Bank (KCLB, Seoul, Korea). Dulbecco’s Modified Eagle Medium (DMEM), penicillin/streptomycin mixture, and fetal bovine serum (FBS) were purchased from GIBCO INC., NY, USA. Fine dust (Urban aerosols—NIES CRM No. 28) was acquired from the National Institute for Environmental Studies (Tsukuba, Ibaraki, Japan). 2′,7′-Dichlorofluorescein diacetate (DCF-DA), 3-(4-5-dimethyl-2yl)-2-5-diphynyltetrasolium bromide (MTT) and dimethyl sulfoxide (DMSO), bovine serum albumin (BSA), Triton-X 100, ethylenediaminetetraacetic acid (EDTA), N-succinyl-Ala-Ala-Pro-Phe p-nitroanilide, and azo dye-impregnated collagen were purchased from Sigma-Aldrich (ST. Louis, MO, USA). cDNA synthesis kit was purchased from ReverTra (Toyobo, Osaka, Japan). BCA protein assay kit, 1-Step transfer buffer, Pierce™ RIPA buffer, protein ladder, PMSF, and propidium iodide (PI) were purchased from Thermo Fisher Scientific (Rockford, IL, USA). Skim milk powder was obtained from BD Difco™ (Sparks, MD, USA). Relevant antibodies were purchased from Cell Signaling Technology Inc. (Beverly, MA, USA) and Santa Cruz Biotechnology Inc. (Dallas, TX, USA). PCR primers were purchased from Bioneer Inc. (Daejeon, South Korea). ELISA kits for human IL-6 and TNF-α were obtained from BioLegend (San Diego, CA, USA). The remaining chemicals and reagents used were of analytical grade. The text continues here.

### 4.2. Isolation and Purification Method of LMF from S. confusum

The LMF used in this study was obtained from one of our previous studies [39]. The isolation and purification method of low-molecular-weight fucoidan from *S. confusum* was described in a previous publication [39]. In brief, the dry powder of *S. confusum* was depigmented by washing with a methanol–chloroform mixture. The depigmented powder was suspended in an ethanol–formaldehyde mixture for 5 h at 40 °C. Then, the powder was washed with ethanol followed by drying. The dried powder was suspended in deionized water at pH 4.5. The suspension was incubated with celluclast for 8 h at 50 °C. The filter and filtrate were incubated in boiling water. After that, 1% CaCl_2_ was added to the filtrate and the supernatant was collected via centrifugation. Then, fucoidan fractions were obtained using a step gradient ethanol precipitation. Four different fractions of fucoidan were obtained based on the molecular weights. The molecular weights were measured with agarose gel electrophoresis and fucoidan was identified by using FTIR analysis, ^1^H NMR analysis, and monosaccharide composition analysis. The lowest-molecular-weight fraction, which was approximately 20 kDa, was used for this study.

### 4.3. HaCaT Keratinocyte-HDF Integrated Cell Culture, FD Stimulation, and LMF Treatment

HaCaT keratinocytes and HDFs purchased from the Korean Cell Line Bank (KCLB, Seoul, Korea, were cultured in a humidified incubator. The temperature inside the incubator was 37 °C with 5% CO_2_. DMEM supplemented with 10% of FBS and 1% penicillin/streptomycin mixture was used to culture HaCaT keratinocytes while the above media mixed with 25% F-12 was using to culture HDFs. HaCaT keratinocytes were sub-cultured once every 3 days and HDFs were sub-cultured once every 5 days at an 80% confluence. The cell culture media in HDFs was replaced with new media after 2 days from the subculture day. HaCaT keratinocytes were seeded in culture dishes with 1 × 10^5^ cells/mL density after the cells reached exponential growth within 3–6 passages and were incubated for 24 h. Following 24 h, the cells were treated with 31.3, 62.5, and 125 μg/mL LMF concentrations as treatment groups. The cells were induced using 150 μg/mL FD for 30 min, the FD was removed by washing with new culture media, and the cells were incubated for 24 h with new media. Then, the media of treatment groups, stimulated groups, and control groups were collected and stored at −80 °C, followed by filtration for further uses as HDF treatment. There were six experimental groups in the HDF culture models. The control group contained HDFs without any treatments. The second group was named ‘Treatment group I’ in the figures and was treated with the media collected from the control group of HaCaT keratinocytes. The third group was named ‘Treatment group II’, which is treated with the FD-stimulated culture media collected from HaCaT keratinocytes. The fourth, fifth, and sixth groups were named ‘Treatment group III, IV, and V’ and were treated with the preconditioned culture media with the treatments of 31.3, 62.5, and 125 μg/mL LMF concentrations, respectively. The HaCaT keratinocytes were seeded in desired culture plates or dishes for predetermined experiments accordingly. HDFs were seeded at a 2 × 10^5^ cells/mL density in culture plates or dishes and incubated for 24 h. Then, the cells were induced by preconditioned HaCaT media for 30 min, the media was replaced with new media, and incubation occurred for the desired time according to each experiment (Figure 7).

### 4.4. Investigation of Cell Viability

The cell viability of HaCaT keratinocytes and HDFs was investigated by using the MTT assay. In brief, HaCaT keratinocytes were seeded in a 96-well plate and incubated for 24 h at 37 °C. Then, the cells were incubated and stimulated with FD, followed by incubation with LMF samples. Then, 15 µL of 5 mg/mL MTT reagent was added to each well after incubation for 24 h. The formazan crystals that formed within 4 h of incubation at 37 °C were dissolved in DMSO for 30 min. Then, the absorbance was measured at 570 nm using a SpectraMax M2 microplate reader. The cell viability of HDFs was evaluated by following the above procedure with slight modifications. In brief, cell culture media in HDF-seeded 96-well plates were replaced with preconditioned media. Then, the media was replaced with new media and incubated for 24 h. Absorbance was measured by following the pre-described method.

### 4.5. Analysis of Intracellular ROS Production

To evaluate intracellular ROS production, a DCF-DA assay was used. In brief, after 24 h of incubation, HaCaT keratinocytes were subjected to FD stimulation following LMF sample treatment. Then, they were allowed to incubate for 1 h with new culture media. DCF-DA was added to each well to measure the intracellular ROS production. Then, the plates were excited at 485 nm and emission was measured at 528 nm by using the SpectraMax M2 microplate reader. Aside from fluorometry, intracellular ROS production was examined by using a CytoFLEX flow cytometer. The HDFs were subjected to preconditioned media and intracellular ROS generation was measured by using the same method described above.

### 4.6. Investigation of Apoptosis

Apoptosis in both HaCaT keratinocytes and HDFs was evaluated by using flow cytometry analysis followed by the propidium iodide (PI) staining method mentioned by Fernando et al. (2017) with slight modifications [41]. Briefly, cells were seeded in a 24-well plate. Then, the sample treatment and stimulation/incubation with preconditioned media were done in the same manner as the procedure mentioned in Section 4.3 above. Then, the cells were incubated for 24 h prior to the analysis. Harvested cells were fixed in 70% ethanol at 4 °C overnight. The cells were treated with 2 mM of ethylenediaminetetraacetic acid (EDTA) after being washed with PBS. After that, the cells were resuspended in 300 µL of PBS–EDTA-containing 0.2 μg/mL RNase and 50 μg/mL propidium iodide for 30 min. The analysis was done using a flow cytometer.

### 4.7. Western Blot Analysis

HaCaT keratinocytes and HDFs were seeded in 6 cm culture dishes for 24 h before the sample treatment/media replace with preconditioned media. Sample treatment/stimulation with FD and stimulation of HDFs with preconditioned media were done according to the method described in Section 4.3. The expression of the MAPK signaling molecules and NF-κB signaling molecules were investigated by using Western blot analysis according to the method described in one of our previous studies [17].

### 4.8. RNA Extraction and RT-PCR Analysis

The mRNA expressions were investigated by using RT-PCR. HDFs were seeded in 6 cm cell culture dishes and all steps mentioned in Section 4.3 were followed. Then, RT-PCR analyses were conducted in the method described by Jayasinghe et al. [17]. In brief, cDNA was synthesized from extracted mRNA by using the ReverTra Ace-α-^®^ cDNA synthesis kit. RT-PCR was performed on the prepared cDNA using the relevant primers. The RT-PCR products were electrophoresed on 1.5% agarose gels with 0.5 µg/mL ethidium bromide before being visualized with a Wisd WUV-L20 UV transilluminator (Daihan Scientific Co., Wonju-si, Korea). The relative intensities of expression levels were measured using ImageJ software (Version 1.52a, US National Institutes of Health, Bethesda, MD, USA), which was then standardized with GAPDH. A list of primer sequences for RT-PCR is mentioned in Table 1.

### 4.9. ELISA Analysis

The HDFs were seeded and treatments were done according to the steps mentioned in Section 4.3. The media from cell cultures were collected into the e-tubes. The production of inflammatory mediators was evaluated using the ELISA assay kits by following the manufacturer’s instructions. In brief, the collected media were incubated in coated plates. Then, detection antibodies were treated, followed by being washed with washing buffer. Stop solution was added to the plate after incubation with streptavidin-HRP. A SpectraMax M2 microplate reader was used to detect the absorbance at 450 nm. Samples were normalized using the standard curves.

### 4.10. Investigation of Intracellular Collagenase and Elastase Activity

HDFs were seeded and incubated in six-well plates at 37 °C with 5% of CO_2_ for 48 h. Then, cells were treated with preconditioned media for 2 h, followed by 24 h incubation with new media. Then, cells were harvested and lysed with 0.1 M Tris-HCl buffer via sonication at a low temperature. The lysing buffer contained 1 mM PMSF and 0.1% Triton-X 100. Following that, a BCA protein assay kit was used to measure the amount of total protein in the lysate. The elastase was measured by mixing 50 µL standardized lysate with 150 µL of 1.015 mM N-succinyl-Ala-Ala-Pro-Phe p-nitroanilide in 10 mM Tris-HCL buffer (pH 8.0). The mixture was gently mixed at 25 °C for 15–20 min and measured the absorbance at 410 nm. The collagenase content was measured by mixing 50 µL standardized lysate with 1.00 mg of azo dye-impregnated collagen in e-tubes and homogenized in 750 μL of 0.1 M Tris-HCl (pH 7.0). The mixture was incubated at 43 °C for 1 h and centrifuged for 10 min. Finally, absorbance was measured at 550 nm.

### 4.11. Statistical Analysis

All statistical analyses of the study were performed using the SPSS software (Version 24.0, Chicago, IL, USA). One-way analysis of variance (ANOVA) followed by Duncan’s multiple range tests were used to evaluate the significant variations among data sets, and data were presented as the mean ± standard error of the mean (SEM). In this study, *p* < 0.05 was considered statistically significant.

## 5. Conclusions

After analyzing all results, it was found that exposure to FD increased the oxidative stress in epidermal cells, thereby increasing apoptosis by up to 467% compared with the control group, which was defined as 100%. Due to changes in the microenvironment of keratinocytes, HDFs also experienced oxidative stress after incubation with preconditioned media, but interestingly, apoptosis in HDFs was insignificant compared with the keratinocytes. FD stimulation significantly increased the phosphorylation of NF-κB/MAPK signaling pathway proteins in keratinocytes, whereas it was dose-dependently downregulated by the LMF treatment. Similarly, the FD-stimulated media treatment moderately increased the phosphorylation of the NF-κB/MAPK signaling pathway proteins in HDFs. Moreover, the media collected from the control group of HaCaT keratinocytes increased the inflammatory cytokines and chemokines, as well as TIMPs and MMPs, in HDF cells at minor levels. Changes in the media of FD-stimulated HaCaT keratinocytes severely affected the HDFs to increase their inflammatory mediators, as well as the mRNA expression of TIMPs and MMPs. Similarly, relative collagenase and elastase activities were increased, whereas TIMPs and MMPs corresponded to ECM degeneration. The preconditioned media protected the HDFs with LMF treatment in a dose-dependent manner. Thereby, the applicability of low-molecular-weight fucoidan isolated from *S. confusum* on the skin to protect not only epidermal cells but also dermal cells were demonstrated. Additionally, the usability of integrated culture models for in vitro studies with correlated cells was suggested.

## Figures and Tables

**Figure 1 marinedrugs-21-00012-f001:**
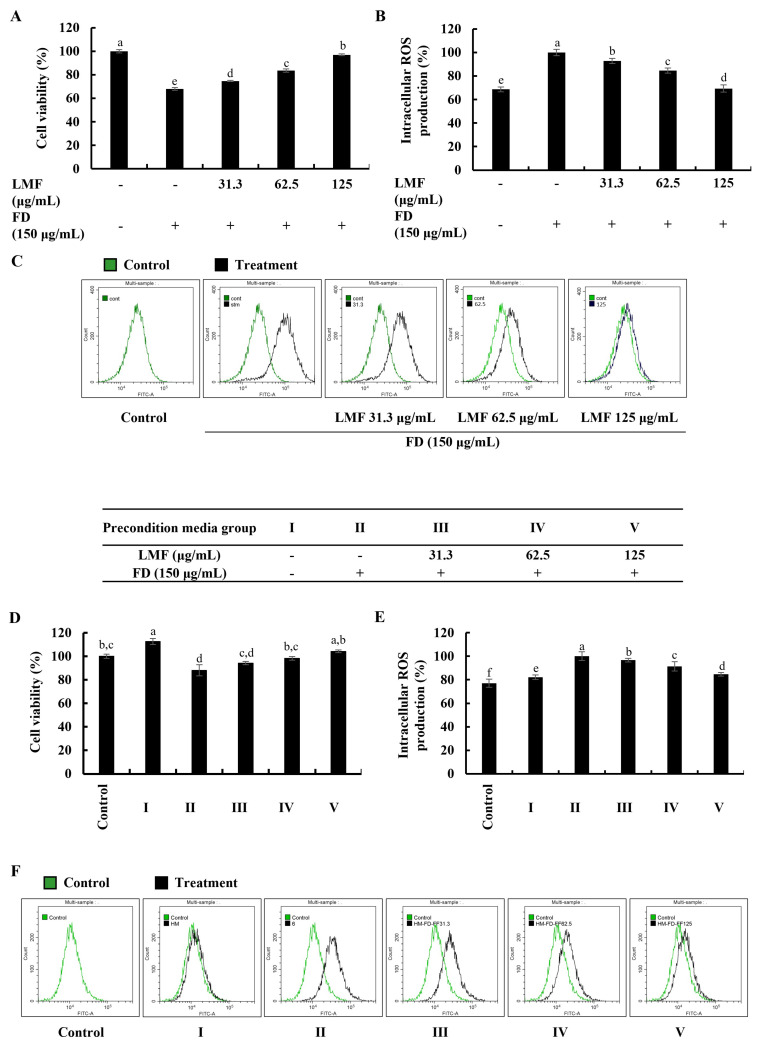
Effect of LMF in FD-stimulated HaCaT keratinocytes and preconditioned media in HDFs in the integrated-culture model. (**A**) Cell viability and the (**B**) fluorometric and (**C**) flow cytometric analyses of intracellular ROS production, with and without FD stimulation in HaCaT keratinocytes. Evaluation of the (**D**) cell viability and the (**E**) fluorometric and (**F**) flow cytometric analyses of intracellular ROS production in HDFs. Preconditioned media from FD-stimulated HaCaT keratinocytes was used to stimulate HDFs for 2 h with or without LMF pre-treatment. Then, 2 h following the stimulation period, the levels of intracellular ROS were assessed. After 24 h, cell viability was measured. All experiments were carried out in triplicate (*n* = 3) and represented as the mean ± SE. Columns in the same graph with different letters are significantly different (*p* < 0.05).

**Figure 2 marinedrugs-21-00012-f002:**
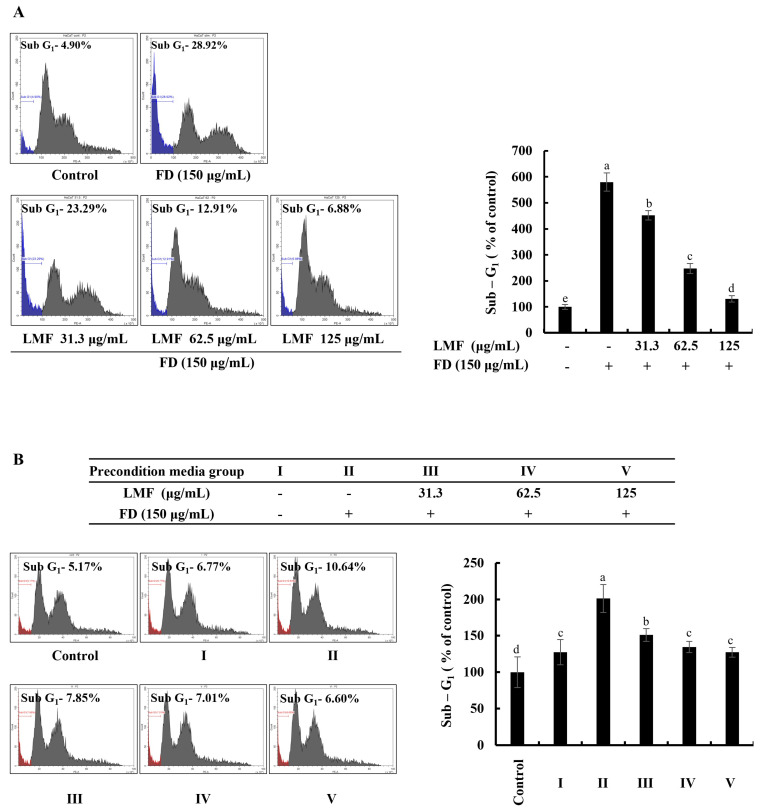
Effect of LMF in FD-stimulated HaCaT keratinocytes and preconditioned media in HDFs on the sub-G_1_ phase of the cell cycle. (**A**) The sub-G_1_ phase of the cell cycle in FD-stimulated HaCaT keratinocytes. (**B**) Evaluation of the sub-G_1_ phase of the cell cycle in preconditioned media-treated HDFs. Evaluations were carried out in triplicates (*n* = 3) and indicated as the mean ± SE. Columns in the same graph with different letters are significantly different (*p* < 0.05).

**Figure 3 marinedrugs-21-00012-f003:**
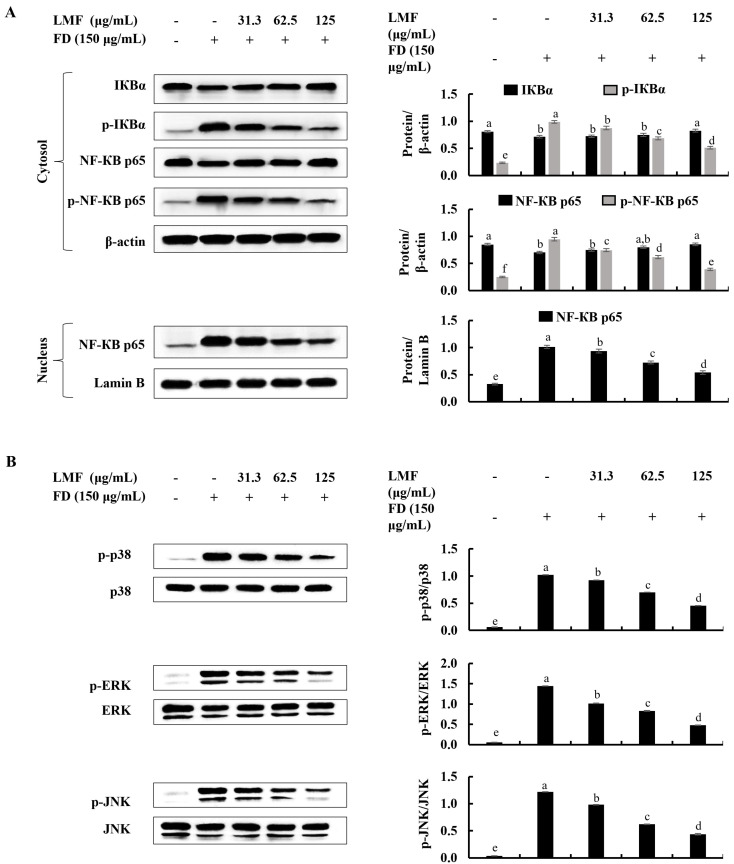
Effect of LMF on NF-κB and MAPK signaling pathways in FD-stimulated HaCaT keratinocytes. (**A**) Western blot analysis of NF-κB and (**B**) MAPK protein expression with and without FD stimulation in HaCaT keratinocytes followed by LMF treatment. Three independent determinations were used to ensure the repeatability of the results (*n* = 3, mean ± SE). Columns in the same graph with different letters were significantly different (*p* < 0.05) for each molecule.

**Figure 4 marinedrugs-21-00012-f004:**
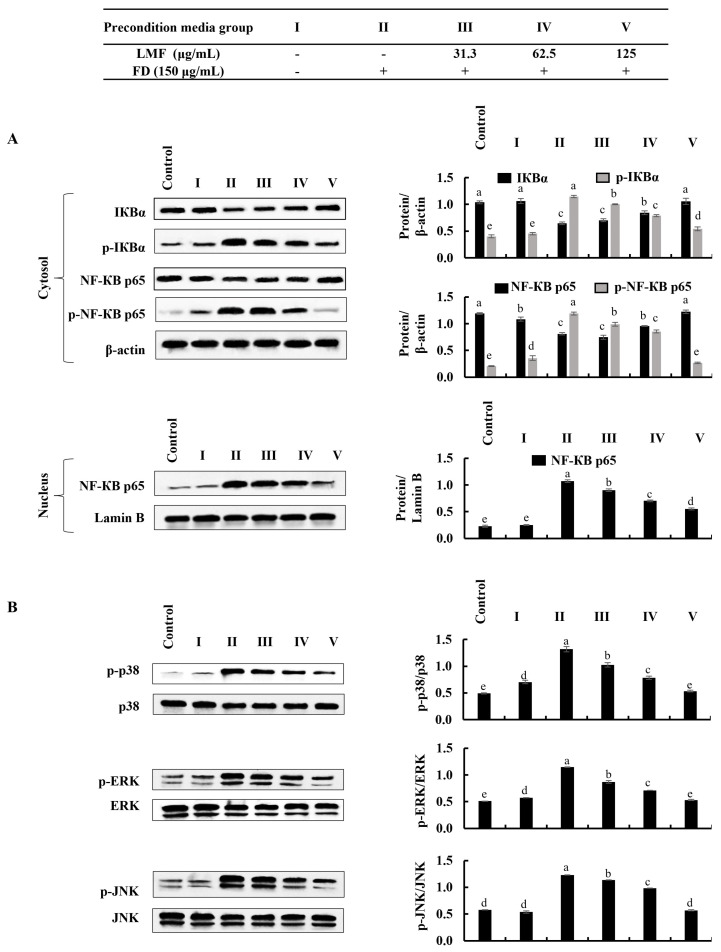
Effect of preconditioned media on NF-κB and MAPK inflammatory signaling in HDFs. Western blot analysis was used to measure the levels of molecular mediators for the (**A**) NF-κB and (**B)** MAPK signaling pathways. Evaluations were performed in triplicates (*n* = 3) and indicated as the mean ± SE. Columns in the same graph with different letters were significantly different (*p* < 0.05) for each molecule.

**Figure 5 marinedrugs-21-00012-f005:**
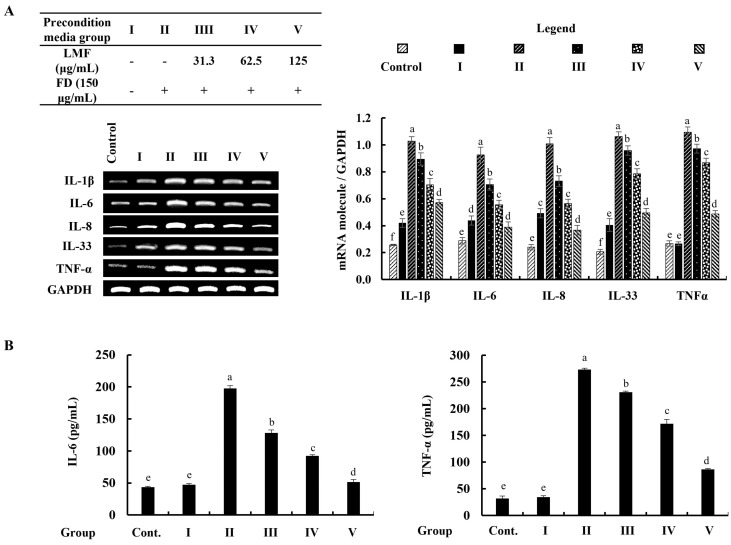
Effect of preconditioned media on the (**A**) mRNA expression of interleukins and TNF-α, and the (**B**) ELISA analysis of IL-6 and TNF-α production in HDFs. Data are represented as the mean ± SE of three independent determinations (*n* = 3). Columns in the graph with different letters were significantly different (*p* < 0.05) for each molecule.

**Figure 6 marinedrugs-21-00012-f006:**
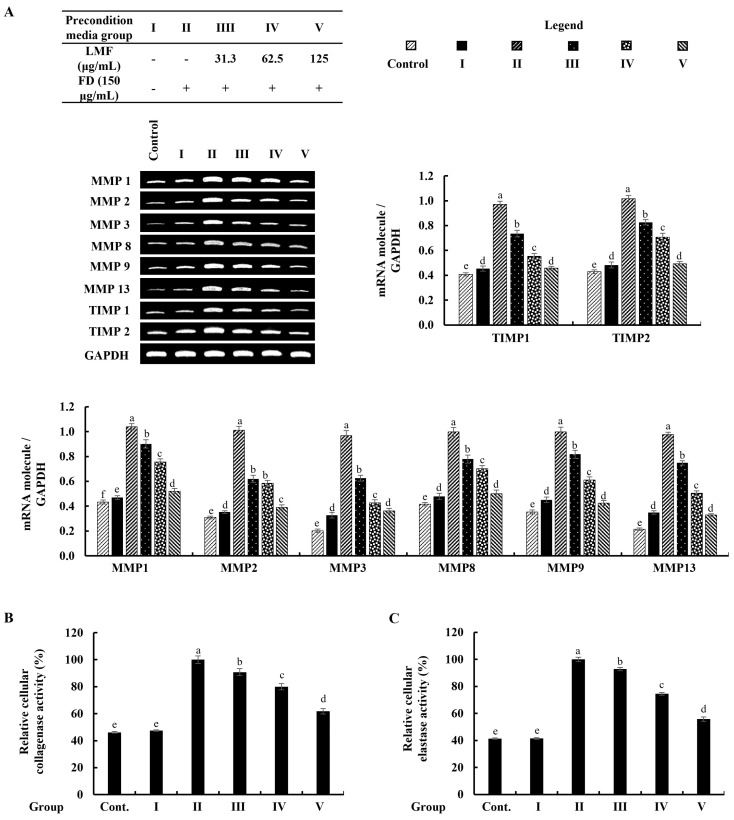
Effect of the preconditioned media on ECM degradation in HDFs. RT-PCR analysis was used to evaluate the (**A**) MMP and TIMP expressions. The activity of (**B**) collagenase and (**C**) elastase was assessed in cell lysates. The values represent data from three distinct experiments (*n* = 3) and are displayed as the mean ± standard error (SE). Columns in the same graph with different letters were significantly different (*p* < 0.05) (in each molecule in (**A**)).

**Figure 7 marinedrugs-21-00012-f007:**
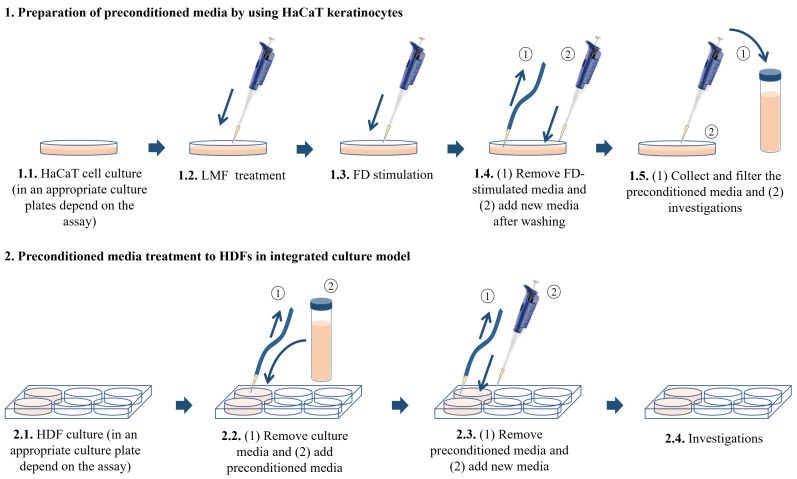
Integrated culture model of HaCaT keratinocytes with HDFs.

**Table 1 marinedrugs-21-00012-t001:** List of primer sequences used for RT-PCR.

Target Gene		Primer Sequence (5′ to 3′ Direction)
IL-1β	Forward	TGT CCT GCG TGT TGA AAG ATG A
	Reverse	CAG GCA GTT GGG CAT TGG TG
IL-6	Forward	GAT GGC TGA AAA AGA TGG ATG C
	Reverse	TGG TTG GGT CAG GGG TGG TT
IL-8	Forward	ACA CTG CGC CAA CAC AGA AAT TA
	Reverse	CAG GCA GTT GGG CAT TGG TG
IL-33	Forward	GAT GAG ATG TCT CGG CTG CTT G
	Reverse	AGC CGT TAC GGA TAT GGT GGT C
TNF-α	Forward	GGC AGT CAG ATC ATC TTC TCG AA
	Reverse	GAA GGC CTA AGG TCC ACT TGT GT
MMP 1	Forward	CTG AAG GTG ATG AAG CAG CC
	Reverse	AGT CCA AGA GAA TGG CCG AG
MMP 2	Forward	GCG ACA AGA AGT ATG GCT TC
	Reverse	TGC CAA GGT CAA TGT CAG GA
MMP 3	Forward	CTC ACA GAC CTG ACT CGG TT
	Reverse	CAC GCC TGA AGG AAG AGA TG
MMP 8	Forward	ATG GAC CAA CAC CTC CGC AA
	Reverse	GTC AAT TGC TTG GAC GCT GC
MMP 9	Forward	CGC AGA CAT CGT CAT CCA GT
	Reverse	GGA TTG GCC TTG GAA GAT GA
MMP 13	Forward	CTA TGG TCC AGG AGA TGA AG
	Reverse	AGA GTC TTG CCT GTA TCC TC
TIMP 1	Forward	TTC TGG CAT CCT GTT GTT GCT
	Reverse	CCT GAT GAC GAG GTC GGA ATT
TIMP 2	Forward	TGG AAA CGA CAT TTA TGG CAA CCC
	Reverse	CTC CAA CGT CCA GCG AGA CC
GAPDH	Forward	CGT CTA GAA AAA CCT GCC AA
	Reverse	TGA AGT CAA AGG AGA CCA CC

## Data Availability

The data are included within the article.

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
