# Peer review of "Fine-Dust-Induced Skin Inflammation: Low-Molecular-Weight Fucoidan Protects Keratinocytes and Underlying Fibroblasts in an Integrated Culture Model"

_marinedrugs, 2022, doi:10.3390/md21010012_

Round 1
Reviewer 1 Report
The article entitled "Fine Dust-induced Skin Inflammation: Low Molecular Weight Fucoidan Protects Keratinocytes and Underlying Fibroblasts in an Integrated Culture Model" is an interesting research article with a lot of novel information. However, this article requires major revisions before consider for publication in Marine drugs.
Major comments
1. Mention about few previous studies related to the Sargassum confusum to strengthen the potential usability of the sample in the study. (In introduction)
2. Rewrite the last paragraph in the introduction with highlighted details about the co-culture model.
3. Please add the DCF-DA FACs results in the 2.1 paragraph which is mentioned in figure 1 to increase clarity.
4. Please keep the consistency of units throughout the manuscript (ex. See Figure 1)
5. Rewrite the figure caption under figure 5 to increase the clarity.
6. Add the characterization details about LMF to the discussion (Like molecular weight distribution, and mono-sugars…)
7. Justify the changes in collagenase and elastase activity by combining the relationship with MMPs in conclusion (Line 456)
8. Mention the usability of the present co-culture model in further studies at the end of the conclusion.
9. Consider removing the word “suggested that” from the conclusion (Line 444) to strengthen the presentation of the findings.
10. Carefully check the capitalization throughout the manuscript (ex. “THE” in the figure 2 figure caption)
Reviewer 2 Report
The manuscript authored by Kirindage and colleagues describes the protective effects of low molecular weight Fucoidan against FD-stimulated inflammation. Overall, the authors are working in an interesting area with the intent to show the isolated polysaccharide can reduce inflammation through NF-κB/MAPK signaling pathway. However, the manuscript suffers from several issues.
1. How about the purity and structural features of LMF? The authors described the structural characterization in Lines 357-360, but no results were presented.
2. What’s the difference between the control group and group I in the figures?
3. Line 366: Is there a reference to this cell model? Was there a concentration-response curve of FD at different times of exposure first established?
4. How about the interaction between FD and LMF? Does the LMF directly bind to FD?
5. Fucoidan has been well-documented for its anti-inflammatory effect, so what’s the merit of LMF in this work? Does the low molecular weight benefit the effect? The authors should compare the effects with the reported ones and make a comprehensive discussion.
Reviewer 3 Report
Dear authors,
The manuscript is well written, presents unpublished data, but needs some minor revisions. Here are my considerations:
1. Authors should explain the difference between control and group I in figures 1, 2, 4, 5 and 6, because it is not clear either in the methodology or in the description of the results.
2. What in the composition of FD would be stimulating ROS and where would FD be binding on keratrinocytes to induce ROS production? Could FD bind to some cell surface molecule or pass through cell membranes and act in the cytosol? How could the authors show this? There is no discussion of this in the manuscript. The authors should punctuate this issue, which is quite relevant. In view of this, how could Fucoidan interfere with this action by the FD?
3. In a previous manuscript, the authors determined the expression of alarmins IL-33, IL-25 and TSLP in TNF-α/IFN-γ-stimulated HaCaT keratinocytes and evaluated the role of fucoidam in this stimulation (reference 17 of this manuscript). Knowing that these alarmins play an extremely important role in the response to skin-damaging agents, like FD, why did not the authors evaluate the expression of TSLP and IL-25, in addition to IL-33, in the current model? It would be important for the authors to measure TSLP, IL-25 and IL-33 by ELISA in this current study and that they establish the relationship of these cytokines with the development of the type 2 inflammatory process that FD can trigger and evaluate whether fucoidan can interfere in this process.
4. Why the authors did not measure cytokines by ELISA, in addition to evaluating their expression by RT-PCR? They did this dosage in a manuscript that is cited in the references of this manuscript (reference 17). It would be important to measure cytokines, as the expression of messenger RNA cannot always lead to protein production. For example, IL1b and IL-33 are produced in the form of pro-IL1b and pro-IL-33, and for them to be produced, activation of the inflammasome complex is required. However, it was not shown in the current study that this molecular complex is being triggered by induction of IL1b and IL-33. Therefore, messenger RNA expression alone is not indicative of cytokine production.
5. The authors must explain in the methodology that they are performing RT-PCR for inflammatory cytokines. They must also include the sequences of the primers used in performing the PCR.
6. Authors must place the composition of the fine dust in the methodology and not in the discussion (lines 251-253). And although it was obtained according to a cited reference, they could briefly describe how it was obtained.
7. Authors should check the way 15 um is written on line 254 and fix it.

Round 2
Reviewer 1 Report
The authors have taken the reviewer's comments into full consideration and the revised version of the manuscript. is well reflected. Therefore, I recommend to consider this manuscript to publish in Marine drugs.
Reviewer 2 Report
The manuscript was well-revised. All of the previous concerns were appropriately solved. Therefore, it can be accepted for publication.